# Chitosan/Montmorillonite Coatings for the Fabrication of Food-Safe Greaseproof Paper

**DOI:** 10.3390/polym13101607

**Published:** 2021-05-16

**Authors:** Kaipeng Wang, Lihong Zhao, Beihai He

**Affiliations:** State Key Laboratory of Pulp and Paper Engineering, South China University of Technology, Guangzhou 510000, China; kaipengwang01@163.com (K.W.); ppebhhe@scut.edu.cn (B.H.)

**Keywords:** chitosan, montmorillonite, coated paper, greaseproof paper

## Abstract

Here, we report a non-toxic method for improving the oil-resistant performance of chitosan coated paper by coating the mixture of chitosan and montmorillonite (MMT) instead of coating chitosan solution only. Through combining MMT into the chitosan coatings, the coated paper exhibited a lower air permeability and enhanced oil resistance under a lower coating load. For coated papers C2.5 and C3 by coating 2.5% (*w/v*) and 3% (*w/v*) chitosan without adding MMT in the chitosan coating, the coating load was 3.76 g/m^2^ and 3.99 g/m^2^, respectively, and the kit rating values were only 7–8/12. Regarding the sample C2M0.1 coated by the mixed solution containing 2% (*w/v*) chitosan and 0.1% (*w/v*) MMT, its coating load was only 3.65 g/m^2^, the paper permeability after coating was reduced to 0.00507 μm/Pa·s, owing to the filling of MMT into the cellulosic fibers network, and the kit rating reached 9/12. Moreover, C2M0.1 showed improved mechanical properties, whereby its tearing resistance was 5.2% and 6.6% higher than that of the uncoated paper in the machine direction and the cross direction, respectively.

## 1. Introduction

Greaseproof paper refers to the paper that can give oil and grease resistance [1]. It is widely used in the packaging of hamburgers, baked bread, biscuits, and other oily goods, etc. [2]. Traditional paper has poor oil resistance due to the presence of porous structure and polar hydroxyl groups in the molecule chains of cellulosic fibers [3]. Therefore, treatment with lamination, low surface energy reagent is often used to solve the problem [4,5]. Synthetic polymers (e.g., polyethylene (PE) and polypropylene (PP)) are commonly applied to laminate paper for nice oil resistance [6,7]. However, the micromolecules of synthetic polymers can migrate from a paper container to foods, which is harmful for human health [8,9]. Using low surface energy chemicals per- and polyfluoroalkyl substrates (PFAS) and silicone oils is another way of improving the oil resistance of paper [3,10]. However, PFAS have a huge threat to human health due to their toxicity and the degradation difficulty [11,12,13]. Additionally, silicone oils beyond a certain limit are also not safe for food contact applications [3].

Moreover, when comparing to the petrochemical-based polymers, biobased polymers are also used to coat paper to achieve oil resistance, owing to their non-toxic, renewable, and biodegradable nature [14,15,16]. As one of biobased polymers, chitosan has also received attention. Dhwani et al. [17] recently reported that the kit rating values of chitosan coated paper and chitosan-graft-sunflower oil coated papers were 12/12 and 8/12, in which the coating loads were 3.6 g/m^2^ and 5.6 g/m^2^, respectively. Syeda et al. [10] showed that the chitosan coated paper had a kit rating value of 12/12 when the coating load was 6.1 g/m^2^, and the chitosan-graft-polydimethylsiloxane coated paper obtained the eighth grade when the coating load was 9.4 g/m^2^. Zhao et al. [18,19] reported that the chitosan coated paper had a kit rating value of 8/12 when the coating load was 9.33 g/m^2^. They further employed chemically modified chitosan in combination with polydimethylsiloxane and castor oil, respectively, and the kit rating values of coated papers were 12/12 and 10/12 when the coating loads were 10.33 g/m^2^ and 4.3 g/m^2^. Kopacic et al. [20] showed that the kit rating values were 6/12 and 5/12 of chitosan coated paper when the coating loads were 6.0 g/m^2^ and 6.1 g/m^2^ for the primary-fiber and secondary-fiber samples. Zhao et al. [21] reported that the chitosan coated paper exhibited a kit rating value of 8/12 when the coating load was 3.63 g/m^2^, they used an aqueous micellar solution of chitosan-graft-polydimethylsiloxane to coat paper substrates, and the kit rating value was 11.7/12 when coating load was only 3.26 g/m^2^. Overall, most of the reports showed that high coating load was often required for a great oil resistance, which would inevitably increase the production cost of the coated paper. Therefore, reducing the coating load of chitosan while maintaining the oil resistance performance has become one focus of the future research. Wang et al. [22] reported that the paper coated by the mixed solution of chitosan and MMT showed better oxygen and water vapor repellencies than those that were coated by chitosan, which provided a new idea for the barrier property enhancement of coatings with chitosan and clay.

So far, there is a lack of research on improving the oil resistance of chitosan coating by mineral powder. Additionally, particularly, the effect of MMT on the oil resistance mechanism of chitosan coating has not been studied. Therefore, in this work, chitosan/MMT blend was used to coat Kraft paper to explore the influence of MMT in chitosan coating on the oil resistance, air permeability, surface texture, and mechanical properties. The present study is beneficial for understanding the principle of oil resistance and further designing excellent oil-resistant paper for food package application.

## 2. Methods

### 2.1. Materials

Chitosan (molecular weight Mn = 50,000–100,000 g/mol, the degree of deacetylation > 95%) was purchased from Macklin Co., Ltd. (Shanghai, China). Montmorillonite was purchased from Yue Wei Co., Ltd. (Beijing, China). Glacial acetic acid (>99.5%) was purchased from Run Jie Co., Ltd. (Shanghai, China). Deionized water was used in all of the experiments described herein. All of the chemicals were used without further purification. Kraft Paper was purchased from Tian He Paper Co., Ltd. (Taian, China).

### 2.2. Preparation of Chitosan Solution and MMT/Chitosan Solution

#### 2.2.1. Chitosan Solution

2% (*w/v*) chitosan solution was prepared by dissolving 2 g of chitosan into 1% (*v/v*) acetic acid to provide a total volume of 100 mL, and this solution was stirred for 24 h at ambient temperature. Further we made 1%, 1.5%, 2.5%, and 3% (*w/v*) chitosan solutions by the same method.

#### 2.2.2. MMT/Chitosan Mixture

Firstly, 20 g MMT was dispersed into 100 mL 1% ((*v/v*) glacial acetic acid solution to prepare 20% (*w/v*) MMT dispersion. Subsequently, the MMT dispersion was added to the chitosan solution with the concentration of 1%, 1.5%, 2%, 2.5%, and 3% (*w/v*), as described above; consequently, the mass ratios of MMT content (dry weight) to chitosan solution were 0.1%, 0.15%, and 0.3% (*w/w*), respectively. Additionally, the prepared mixture was magnetically stirred (Bo Ke Co., Ltd. Jinan, China) for 4 h, and then kept it static for 30 min. to eliminate foam, so that the MMT/chitosan solutions with different ratios were obtained.

#### 2.2.3. Coating Preparation

The coatings were applied on the papers surface using the rod coating method with rod number 8. For each coating, a 5.0 mL solution was applied on an 11 × 11 cm^2^ Kraft paper and then dried for 24 h at room temperature. Additionally, then, these coated papers were preconditioned at 23 °C under 50% relative humidity before performing further studies.

### 2.3. Coating Load, Thickness

The coated load was calculated using Equation (1) and expressed in g/m^2^. The thickness of paper was tested with the L&W micrometer (L&W, Stockholm, Sweden) at five different places on each sample. The final reported value corresponded to an average of five measurements.
(1)$$\mathrm{Coating}\;\mathrm{load}\;\left(g/m²\right)=\frac{\mathrm{Coating}\;\mathrm{paper}\;\left(g\right)-\;\mathrm{Unmodified}\;\mathrm{paper}\;\left(g\right)}{\mathrm{Coating}\;\mathrm{paper}\;\left(m²\right)}$$

### 2.4. Grease Resistance

Uncoated paper and coated paper were tested to evaluate their grease resistance following the TAPPI standard T559pm-96 protocol. 12 kit rating solutions were prepared by mixing castor oil, heptane, and toluene in various ratios, and they were numbered from 1 to 12, representing oil resistance from poor to excellent. During each test, one kit rating solution was added dropwise onto the paper for 15 s and then wiped away. If no oil mark was left on the paper surface after 15 s, the kit rating number was considered to be passed, and a higher numbered solution was then tested until the surface failed the test. The value reported herein for each sample corresponded to the average on three records.

### 2.5. Experimental Design and Statistical Study

Response surface methodology (RSM) was employed to determine the optimum concentrations of coating solutions of chitosan and MMT to obtain the oil/grease resistance. The central composite design (CCD) with three levels and two factors, including the concentrations of the chitosan (X_1_) and the MMT (X_2_), was performed using JMP Statistical Software (SAS Institute Inc., Cary, NC, USA). Preliminary experiments were performed to determine a reasonable range of the two independent variables. The concentration of the chitosan solutions varied within the range of 1–3% (*w/v*), in which the best oil/grease resistance was obtained. Meanwhile, the concentration of MMT varied between 0% (*w/v*) and 0.3% (*w/v*). The results from the coating load and oil resistance tests were selected as responses, and were then fit into a second-order polynomial regression Equation (2) based on the least-squares-fit method.
(2)$$Y=b_{0}+b_{1}\left(X_{1}-2\right)+b_{2}\frac{X_{2}-0.15}{0.15}+b_{11}{\left(X_{1}-2\right)}^{2}+b_{22}{\left(\frac{X_{2}-0.15}{0.15}\right)}^{2}$$
where X_i_ is an independent variable and Y is the dependent variable, while bi and bij are regression coefficients. The statistical significance of each coefficient was studied using analysis of variance based on the *p*-values, and the accuracy of the model was evaluated based on the R^2^ value.

### 2.6. Grease Permeability (GP)

The Grease Permeability of the uncoated and coated papers was tested using TAPPI standard T454om-10. Briefly, each specimen was placed on a white sheet of paper. Five grams of dry sand was placed into a 25 mm tube and the tube was then put onto the test specimen to obtain a heap and 1.1 mL of red-dyed turpentine was added dropwise to the heap of sand. The time duration was recorded in seconds until the first red turpentine permeation appeared on the white paper under the sample. A length of 1800 s corresponds to a high penetration resistance to fats and oils, and the higher grease resistant paper was the specimen with a longer time (1800^+^ s) [23].

### 2.7. Air Permeability Analysis

The Bendtsen air permeability of paper samples was determined according to the standard GB/T458-2008 in the L&W Air Permeance Tester (L&W, Stockholm, Sweden). The measurement area was 10 cm^2^. Five measurements were taken on each sample.

### 2.8. Scanning Electron Microscopy (SEM)

The micrographs of unmodified and coated papers were recorded with a scanning electron microscopy (JEOL 6400 SEM, JEOL Ltd., Tokyo, Japan). The samples were prepared for SEM characterization by mounting them onto aluminum stubs using a carbon double-sided tape and then spraying with gold.

### 2.9. Contact Angles (CAs) 

The water and oil (castor oil) contact angles (CAs) of the uncoated paper, coated papers were measured with Contact Angle Surface Analyzer (Model OCA20, from Dataphysics Instruments GmbH, Filderstadt, Germany). Droplets testing liquid with volume of approximately 5 μL were placed onto the paper specimen and the values were recorded after 30 s and 5 min. Five replicates were recorded for each sample. 

### 2.10. Surface Roughness

The surface roughness was measured by the 3D Optical Profiler (Rtec-Instruments, San Jose, CA, USA) and the magnification was 20 times. Three measurements were taken on each sample and the mean values were reported.

### 2.11. Thermogravimetric Analysis (TGA)

The TGA data of the uncoated and coated paper were measured with the thermal gravimetric analyzer (NETZSCH-Gerätebau GmbH, Selbu, Germany). Samples weighing around 10–20 mg were heated under an N_2_ atmosphere at a flow rate of 20 mL/min. from 40 °C to 600 °C at a constant heating rate of 10 °C/min. The derivative thermogravimetric (DTG) curve was plotted as a first derivative of the TGA curve.

### 2.12. Mechanical Properties

Tensile strength (TS) tests were performed via the TAPPI standard T494 protocol with the L&W tensile tester (L&W, Stockholm, Sweden). A 15 mm × 100 mm dimension specimen was made from the paper. The tensile strength was calculated by dividing the maximum tensile force by the width of the specimen. 

A 75 mm × 63 mm dimension specimen was made for Internal tearing resistance (ITR) test with the L&W internal tearing tester (L&W, Stockholm, Sweden) following the TAPPI T414.

## 3. Results and Discussion

### 3.1. RSM Results

Table 1 lists the CCD experimental design and corresponding dependent variables. Table 2 contains the estimated coefficients of each model, R^2^ values of 0.93 and 0.94 response for Y_1_ and Y_2_, and both are greater than 90%, serving high precise consequences of the model as compared to the experimental results. Figure 1 shows the surface profiles with the contours of two dependent variables. The results revealed that the coating load played a critical role in improving the oil resistance of the obtained coated paper, and the MMT contributed to improving the oil resistance of chitosan coating. The optimum formulation was determined as: (1) X_1_ = 2, X_2_ = 0.1 (with a desirability of 0.65). Therefore, 2.0% chitosan and 0.1% MMT were chosen as the optimum formulation.

### 3.2. Coating Load, and Thickness of the Paper

The coating load and thickness of the coated and uncoated papers were tested and are shown in Table 3. Based on the thickness of the uncoated paper 102 μm and the coated papers 106–110 μm, the coating thickness covering the paper substrate was calculated out to be 4–8 μm. Additionally, the coating loads ranged from 3.45 to 4.49 g/m^2^. More remarkably, the coated paper sample C2M0.1 had a coating load of 3.65 g/m^2^.

### 3.3. The Oil Resistance and Grease Permeability of Paper

Kit rating values were tested to explore the oil resistance of paper (Figure 2a). The kit rating value of uncoated paper was 0/12, which indicated the lack of oil resistance. While the papers coated with chitosan only (e.g., C2, C2.5, C3) showed an improved kit rating, and with the increase of chitosan concentration, the kit rating increased gradually. Additionally, the addition of MMT improved the oil resistance of chitosan coating until the content of MMT reached up to 0.3%. Moreover, the C2M0.1, C2.5M0.1, and C3M0.1, in which combined chitosan with MMT, showed a higher kit rating of 9/12. Moreover, it was noteworthy that the kit rating value of C2M0.1, with only 0.1% MMT employed, was even higher than C2.5 and C3 (kit rating values were 7/12 and 8/12, respectively), although its coating load was lower than C2.5 and C3.

At the same time, we tested the grease permeability performance of the paper (Figure 2b). Like the results in kit rating, the uncoated paper showed the lowest value for grease resistance (only 3 s), indicating a poor repellence to the grease permeability. Chitosan coated paper showed great grease permeability up to (205–246.6 s) with one layer of chitosan applied (e.g., C2, C2.5, C3). Likewise, the grease resistance first increased and began to decrease at the addition of MMT up to 0.3% (e.g., C2M0.3, C2.5M0.3). Furthermore, through the function of MMT, the grease permeability of C2M0.1 attained to 290.8 s, was even better than C2.5 and C3 with higher chitosan content, and it was close to C3M0.3. The results also indicated that the addition of MMT significantly improved the grease permeability of chitosan coating.

### 3.4. The Air Permeability of Paper

The air permeability of paper samples was also determined to further explore the reason for grease resistance. According to the results shown in Figure 2c, corresponding to its poor oil resistance, the air permeability of uncoated paper was 458.7 μm/Pa·s, it was attributed to the porous structure of the paper (Figure 3a). The application of the chitosan coating onto paper resulted in a reduced air permeability (e.g., C2, C2.5, C3), owing to the chitosan masking on the pores (Figure 3b,d,e), and correspondingly the coated paper showed better oil resistance performance (Figure 2a,b). However, through combining with MMT, C2M0.1 showed a lower air permeability when compared with C2.5 and C3 (Figure 2c), even if it owned the lowest coating load (only 3.65 g/m^2^). It was because the cooperation of chitosan with MMT better filled the pores of the cellulosic fibers network and formed a continuous thin film over the surface [22,24]. However, MMT aggregated and formed block structure (e.g., C2M0.3) over the coating surface while the content of MMT continued to increase (Figure 3f), which was the reason why C2M0.3 had bad oil resistance and high air permeability (Figure 2c).

### 3.5. The Oil Contact Angles (OCAs) and Water Contact Angles (WCAs) of Paper

The wettability of the papers was explored via OCAs and WCAs, as shown in Figure 4. The OCA for the uncoated paper decreased by 11.6° after the oil droplet was absorbed for 5 min. (Figure 4a), and it left a grease spot on the paper surface. Meanwhile, the OCAs of chitosan-coated papers (e.g., C2, C2.5, C3) decreased by 6.4–6.8°. Surprisingly, the reduction in OCA from C2M0.1 with low coating load was only 4.1°, which was the least decline among all of the samples without leaving any grease mark behind, reflecting its good oil resistance stability. In contrast, the reductions in OCAs of C2M0.3, C2.5M0.3, and C3M0.3 was 8.5°, 7.5°, and 9.8° respectively, owing to the aggregation out of the excessive addition of MMT and, consequently, the rougher surface of the coatings (Table 4). This result is consistent with the Wenzel roughness equation, when the equilibrium contact angle is less than 90°, the higher the surface roughness is, the smaller the OCA is [25].

The WCA of the uncoated paper decreased by 24.6° due to the lack of water resistance capability, while the WCAs of coated papers decreased by 10.4–16.2° and showed better water resistance stability (Figure 4b). With the loading increase of MMT, the WCAs of the coated paper series C2, C2.5, and C3 increased, respectively, due to the increase of surface roughness of coatings (Table 4). According to the Wenzel roughness equation, when the equilibrium contact angle was more than 90°, the higher the surface roughness, the larger the WCA. Accordingly, the hydrophobic property for the chitosan coatings improved with roughness increased.

### 3.6. TGA and DGA of Paper and Coating

Thermogravimetric analysis (TGA) was performed to investigate the thermal stability of paper (Figure 5). There were no significant changes in the TGA plots between the coated and uncoated papers because of the low coating loads (Figure 5a). Therefore, free-standing films of chitosan and chitosan/MMT were prepared to gain insight into the weight loss of chitosan coating and chitosan/MMT coating (Figure 5c). The weight loss below 120 °C corresponded to the evaporation of moisture from the paper substrate. The paper substrate decomposed between 292.6 °C and 353.1 °C. The chitosan film showed three weight loss regions. The first weight loss occurred below 160 °C due to the evaporation of water. The second weight loss occurred between 160 °C and 220 °C, and it was likely due to the removal of the acetic acid that had been used for chitosan dissolution. Finally, the weight loss occurred between 220 °C and 400 °C was due to the degradation of the molecular chains of chitosan [26,27]. The thermal stability of chitosan/MMT nanocomposites decreased systematically with clay increasing in general. Our results were similar to the report of Asira [28,29]. In summary, it indicated that the coated paper was thermally stable and, hence, useful for high-temperature applications.

### 3.7. The TS and ITR of Paper

The mechanical properties of paper samples were evaluated (Figure 6). The TS values of the coated papers increased with the increasing of coating loads. The TS of C2M0.1 with coating load 3.65 g/m^2^ was 2% higher in the machine direction and 1% higher in the cross direction than that of the uncoated paper. The variation of the ITR values was similar to the TS values of the coated papers. It was observed that the ITR of C2M0.1 was 5.2% and 6.6% higher than that of the uncoated paper in the machine direction and the cross direction, respectively.

## 4. Conclusions

In summary, the MMT was successfully used to strengthen the oil resistance performance of chitosan coating on Kraft paper. The addition of MMT into the chitosan coating modified the surface roughness and it significantly decreased the air permeability of coated paper, resulting in an advanced oil resistance under a lower coating load. In addition, the coated paper also showed a comparative thermal stability and advanced mechanical properties. This work provides a novel approach for research into the preparation of highly oil-repellent coated paper with low coating volume, and it would offer some economic and environmental benefits that are related to packaging and nonpackaging sectors.

## Figures and Tables

**Figure 1 polymers-13-01607-f001:**
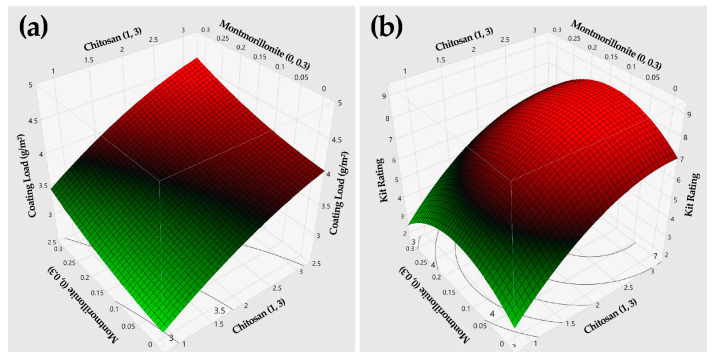
3D surface profiles for (**a**) the coating load (g/m^2^) and (**b**) the kit rating.

**Figure 2 polymers-13-01607-f002:**
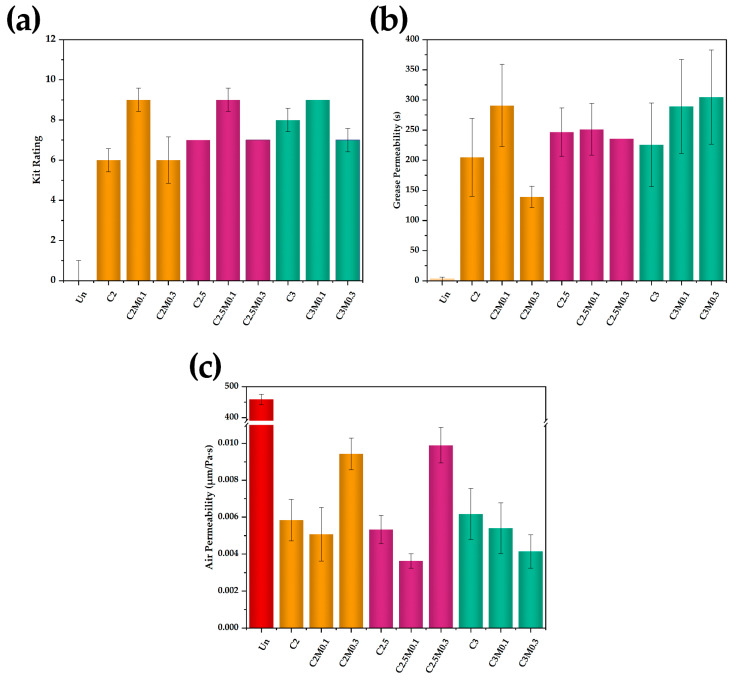
(**a**) Kit rating, (**b**) grease permeability, and (**c**) air permeability (μm/Pa·s) of the uncoated and coated papers.

**Figure 3 polymers-13-01607-f003:**
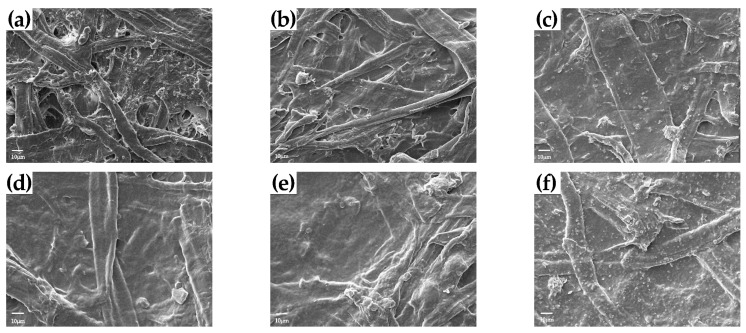
SEM images of the uncoated paper Un (**a**), C2 (**b**), C2M0.1 (**c**), C2.5 (**d**), C3 (**e**), and C2M0.3 (**f**).

**Figure 4 polymers-13-01607-f004:**
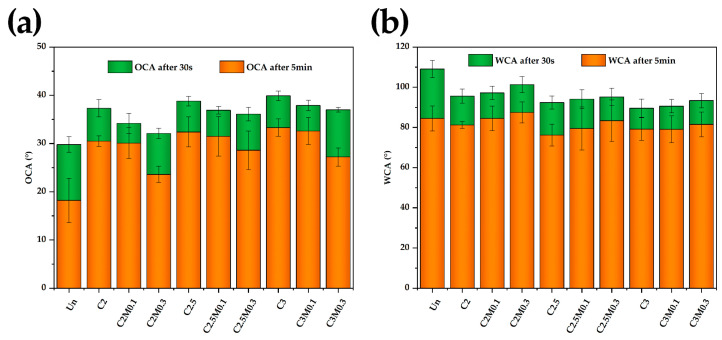
CAs of the uncoated and coated papers to (**a**) castor oil and (**b**) water.

**Figure 5 polymers-13-01607-f005:**
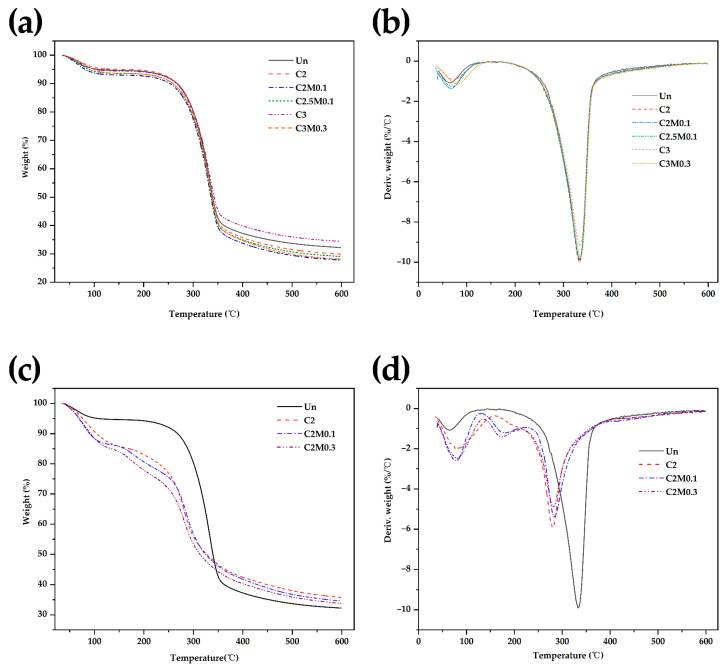
TGA (**a**) and DTG (**b**) plots of the uncoated paper (Un) and coated papers (C2, C2M0.1, C2.5M0.1, C3, C3M0.3). Additionally, TGA (**c**) and DTG (**d**) plots of the uncoated paper (Un), chitosan coating (C2), and mixture coatings (C2M0.1, C2M0.3).

**Figure 6 polymers-13-01607-f006:**
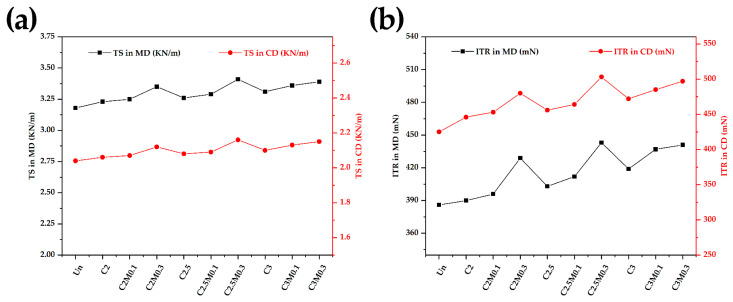
The TS (**a**) and ITR (**b**) of the uncoated paper and coated papers in the machine direction (MD) and cross direction (CD).

**Table 1 polymers-13-01607-t001:** The CCD design of experiment and experimental responses.

	Variables		Responses	
Sample No.	X1 ^a^ (%)	X2 ^b^ (%)	Y_1_ (Coating Load) (g/m^2^)	Y_2_ (Kit Rating)
1	2	0.15	3.81	8
2	1	0.15	3.11	5
3	2	0.15	3.65	8
4	1	0.3	3.50	4
5	3	0	3.99	8
6	1	0	3.17	4
7	3	0.3	4.42	7
8	3	0.15	4.21	8
9	2	0.3	4.18	6
10	2	0	3.45	6
11	2	0.15	3.93	7
12	2	0.15	3.78	8

^a^ The content of the chitosan, ^b^ The content of the MMT.

**Table 2 polymers-13-01607-t002:** Regression coefficients of the second-order polynomial model.

Term	Estimate	Std. Error	*p*-Value
Coating Load			
b_0_	3.78	0.06	<0.0001 *
b_1_	0.47	0.05	<0.0001 *
b_2_	0.25	0.05	0.0017 *
b_11_	−0.09	0.08	0.2769
b_22_	0.07	0.08	0.4079
Kit Rating			
b_0_	7.58	0.23	<0.0001 *
b_1_	1.67	0.21	<0.0001 *
b_2_	−0.17	0.21	0.4512
b_11_	−0.75	0.31	0.0479 *
b_22_	−1.25	0.31	0.0053 *

* Indicated that the coefficient was significant at a 95% confidential level (*p*-value < 0.05).

**Table 3 polymers-13-01607-t003:** The coating load and thickness of the paper prepared in this study ^α^.

Sample Name	Coating Load (g/m^2^)	Material Thickness (μm)
Un	/	102.1 ± 2.4
C2	3.45 ± 0.03	106.0 ± 2.4
C2M0.1	3.65 ± 0.2	106.6 ± 1.9
C2M0.3	4.18 ± 0.22	108.1 ± 2.3
C2.5	3.76 ± 0.11	107.1 ± 2.5
C2.5M0.1	3.94 ± 0.09	108.2 ± 1.7
C2.5M0.3	4.49 ± 0.17	107.1 ± 1.9
C3	3.99 ± 0.19	109.1 ± 3.1
C3M0.1	4.26 ± 0.15	110.0 ± 2.8
C3M0.3	4.42 ± 0.17	109.6 ± 1.6

^α^ Un represents uncoated paper, C represents chitosan, M represents MMT, while the number denotes the concentration in (*w/v*). For example, C2M0.1 represents the coated paper that is coated with the mixed coating solution consisting of 2% (*w/v*) chitosan and 0.1% (*w/v*) MMT.

**Table 4 polymers-13-01607-t004:** The roughness average (Ra) of papers.

Different. Papers	Un	C2	C2M0.1	C2M0.3	C2.5	C2.5M0.1	C2.5M0.3	C3	C3M0.1	C3M0.3
Ra (μm)	7.06	5.49	5.57	5.85	5.26	5.42	5.74	5.09	5.38	5.52

## Data Availability

The data presented in this study are available on request from the corresponding author.

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
