# Peer review of "Chitosan/Montmorillonite Coatings for the Fabrication of Food-Safe Greaseproof Paper"

_polymers, 2021, doi:10.3390/polym13101607_

Round 1

Reviewer 1 Report

Polymers-1203024-peer-review-v1-1

 Manuscript number Polymers-1203024-peer-review-v1-1

Chitosan/Montmorillonite Coatings for the Fabrication of Food-Safe Greaseproof Paper” presents a non-toxic method to improve the oil-resistant performance of chitosan coated paper. The authors used coating chitosan and montmorillonite (MMT) instead of coating chitosan only. In this case, the coated paper exhibited lower air permeability and an enhanced oil resistance under a lower coating load.  The paper permeability after coating was reduced to 0.00507 μm/Pa·s due to the filling of MMT into the cellulosic fibers network. Moreover, that sample showed improved mechanical properties.

The manuscript should have some MINOR revisions before publishing.

Here are some questions and advices for the authors:

-Why you used   the rod coating method?

-The quality of the figures can be improved,

-The english also can be improved.

If the manuscript would have some minor revision before publishing, it will be interesting for the readers of the Polymers.

Author Response

尊敬的评论者1,

请参阅附件。

Reviewer 2 Report

The submitted manuscript presents an interesting topic on packaging materials with an enhanced oil-resistance by using of environmentally friendly materials. The introductory part encompasses the sufficient information on the concerning topic with the relevant literature sources. 

My comments follow:

English language editing is recommended, since I find some sentences not clearly formulated. Unfortunately, I miss the page and line numbering, which makes better specification more difficult.

  • Chapter 2.6. “Briefly, Place each specimen” could be rewritten as “briefly, each specimen was placed on…”.
  • Chapter 3.1. “Table 2 was the estimated coefficients..” could be changed to “Table 2 contains the estimated coefficients…”.
  • In the Chapter 3.3. “While among the coated paper…” seems to be rephrased.
  • In the Chapter 3.4. the sentence “It was because that, the…” does not seem to be grammatically OK.
  •  

I suggest to describe the procedure of MMT-chitosan mixtures preparation in more detail in the appropriate chapter (2.2.2.),  or at least the reference to the sample designation (mentioned in Table 3) should be included. The phrase “Chitosan Solutions” should not be with capital letters.

Fig. 2C should be edited so that the inset graph (legend and axes description) is more readable.

Author Response

Dear Reviewer 2,
